# How do older women perceive the occupations and activities within their maternal role? Findings from an exploratory survey

**Ruth Maman, Debbie Rand** °, **Michal Avrech Bar**°

Department of Occupational Therapy, School of Health Professions, Sackler Faculty of Medicine, Tel Aviv University, Tel Aviv-Yafo, Israel

° These authors contributed equally to this work.
\* michaavr@tauex.tau.ac.il

## Abstract

### Background

Participation in meaningful everyday occupations and life-roles is crucial to the health and wellbeing of older adults. However, little is known regarding meaningful life-roles of older women. Although the maternal-role remains meaningful to women throughout their life, previous literature focused on earlier stages of motherhood.

### Aims

To describe the occupations and perceptions within the maternal-role of older women.

### Materials and method

An online survey was distributed via social media. It included closed and open-ended questions regarding the engagement and relatedness of occupations to the maternal-role; and the perceptions of older women towards their maternal-role. Quantitative data was analyzed using descriptive statistics, and thematic analysis was used to analyze data from open-ended questions.

### Results

The survey was answered by 317 community-dwelling older mothers (aged 65–87). High frequency of engagement and relatedness of occupations to the maternal-role were found. Most participants perceived the maternal-role as a never-ending and evolving life role. Seven categories, describing both 'doing' and 'being' aspects of the maternal-role, were identified.

### Conclusion

The maternal-role is meaningful to older women. It continues to develop over time, and includes new occupations which have not been central at earlier stages of motherhood.

**Data Availability Statement:** The data set is available to other researchers for replication purposes, and was uploaded as supporting materials

**Funding:** The authors received no specific funding for this work.

**Competing interests:** The authors have declared that no competing interests exist.

## Significance

These findings have significant implications for healthcare professionals striving to promote healthy aging by enhancing the participation of older women in meaningful occupations. Further research is needed to broaden the understanding of the unique characteristics of the maternal-role at older age.

## Introduction

Participation in meaningful life roles and occupations is a central factor of healthy aging [1–4]. In contrast to previous perspectives of health and aging which focused on the absence of illness, current definitions emphasize the importance of enabling people to be and do what they value, while maintaining relationships and contributing to society [5]. In accordance, participation of older adults in various activities and occupations, such as work, social and leisure activities, has been found to be positively linked with health and wellbeing [6–9]. Occupations are the daily activities that people need, want or are expected to do in various contexts, such as engaging in leisure activities with family members or shopping in the community [10]. The engagement in occupations carries personalized meaning, which can vary among groups of age, culture and gender [11]. Older women comprise a large and growing part of the global population, and therefore, fostering their health and wellbeing is a central challenge worldwide [12].

Although the life expectancy of women is longer than men [13], older women frequently face compromised health and wellbeing in the remaining years of their life [14–16]. Despite the wide research regarding the benefits of participation on the health of older adults, a striking paucity was found in research that focuses on the occupations and life-roles that are meaningful to older women [17, 18]. A common perception among mothers is that being a mother is a life-long role [19–21]. This suggests that older women who are mothers would perceive their maternal role as a central life-role, and hence would be motivated to participate in meaningful maternal occupations which may promote their health and wellbeing. However, since most of the research regarding motherhood is based on younger mothers in earlier stages of raising children, it is unclear what occupations and activities comprise the maternal role at older age, once the children are grown-up.

Being a mother is a common and desired life-role among adult women [22, 23]. Mothers often prioritize the maternal role over other life roles, such as work or community roles, and experience it as a significant part of their self-identity [24–26]. The maternal role of younger mothers includes participation in various occupations and daily activities, such as preparing meals and driving children to school [27, 28]. The specific occupations that are ascribed to the maternal role vary between women and are widely influenced by social, cultural and personal contexts [29, 30].

The unique occupational choices made by mothers within the maternal role are influenced by personal values and perspectives. Hibbs and Getzen [31] described that mothers continue to engage in mothering-related occupations, even when their children become independent adults and form their own immediate families. In a scoping review of the maternal role at older age, that we conducted, and included 14 articles [32], we also found that older mothers continued to provide various means of support to their adult children (instrumental and emotional among others) and also engaged in various social activities together with them. The older mother's perceptions regarding their relationships with their adult children is an important topic that arose from the review. However, these findings were elicited from scarce

research, which did not originally aim to describe the occupations within the maternal role of older mothers.

According to the Occupational Therapy Practice Framework (OTPF), activities within the maternal role can be identified in two occupations: social participation, which include the engagement in activities within familial roles; and Instrumental Activities of Daily Living (IADL), which include activities of caregiving and supervising for others [10]. However, these categories are too general to provide specific information regarding the unique maternal role activities of older women. Adult (healthy) children do not usually require daily caregiving, and the general category of social participation does not inform which activities are specific to the participation of older mothers with their adult children. In addition, the broad aspects of 'being', which relate to the way mothers feel about what they do and the sense of who they are as occupational human beings [33, 34], might not be fully represented by these categories. The idea that the traditional categorization of occupations is not applicable for older adults was previously argued, since these categories do not encompass the subjective meaning and priorities which lead older adults to choose and engage in occupations [23, 35–37]. This, together with the scarce literature of the maternal role at older age, and the fact that tools to characterize mothering occupations of older women are not available, creates vagueness regarding the occupations within the maternal role of older women. Understanding the unique characteristics of the maternal role at older age is important to healthcare providers, and particularly to occupational therapist practitioners, who hold an important role in the promotion of health and wellbeing of older adults [10]. This information of healthy and independent older mothers may be used in the future to develop health promotion or intervention programs that are tailored specifically for older mothers with disabilities. Therefore, the aim of this exploratory study was to describe occupations and daily activities of older mothers; their perceived relatedness of these activities to the maternal role; and their perceptions towards the maternal role.

## Materials and methods

### Design

This exploratory cross-sectional study utilized an online survey to collect detailed information from older mothers. The survey included mostly closed-ended self-report questions regarding the maternal role at older age. Two open-ended questions were included as well. The ethics committee at Tel Aviv University granted approval prior to recruitment (approval # 0001502–2, July 15 th, 2020).

### Participants

Older women were recruited by advertisement in relevant online social media and by snowball sampling, using the following inclusion criteria: women aged 65 years or older, who independently live in the community and have at least one (living and healthy) child. Prior to answering the online survey, all women indicated digital informed consent to participate. Only then, could participants proceed to answer the survey.

### Tools

The online survey included obtaining information regarding:

**Occupations within the maternal role at older age**–Participants were asked to rate the *relatedness* of each occupation to their maternal role using Likert scales (0-not related, 1-weakly related, 2- moderately related, 3- strongly related) and the *frequency* of their engagement in each of the occupations (never, at least once a year, at least one a month, at least once

a week). Since a standard tool for assessing maternal occupations of older women was not found, the questions were based on the scoping review of the maternal role at older age [32] as well as on relevant aging and motherhood literature [For example: 17, 37–39, 42]. The following occupations were included: passing on family legacy to the next generation, such as sharing life experience with younger family members (generativity); helping family members keep in touch with each another, for example, planning and organizing family get-togethers, celebrating holidays together (kinkeeping); assisting children with their daily chores, such as preparing meals and taking care of the grandchildren (assisting in IADL); providing concrete support or advice in various fields, for example, sharing recipes or household maintenance tips (guidance); sharing everyday occurrences and experiences with children (sharing); providing support such as listening and sharing experiences when children experience challenges (emotional support); being concerned regarding the wellbeing of children (caring); providing financial assistance, such as help with paying rent or college tuition (financial support).

### Perceptions of the maternal role at older age

Eight statements were included, representing possible perceptions regarding the maternal role at older age. For example, 'Being a mother is a life-time role', 'The mothers' role is over when children leave home or when grandchildren are born' and 'My children are independent but they sometimes need my help or presence'. These statements were also based on relevant literature regarding the maternal role and aging [For example: 17, 40–42].

Participants were asked to rate each statement as 'disagree, partially agree, or agree'.

In addition, two open-ended questions were presented and inquired regarding 1) additional activities that the mothers perform within their maternal-role, and 2) other relevant information they can share regarding the maternal-role in older age.

**Socio-demographic information** such as age, years of education, marital status, and number of children was collected.

### Data analysis

Descriptive statistics were used to describe the sample and the data yielded from the closed-ended questions, by using SPSS Statistics 25' software. The information collected from responses to the open-ended questions was analyzed in two steps using thematic analysis [43]: At first, the information was coded by labels, then the labels were grouped to identify collective themes that arose from the participants answers.

## Results

Three hundred and seventeen older mothers participated in the survey. They were aged 65 to 87 years old [mean (SD) age—69.7 (4.2)]. Participants had 1 to 7 children, with a mean (SD) of 3.0 (1.0) children. Most participants were born in Israel, married or in a relationship, live with a spouse, in their own home in urban settings (see Table 1).

### Frequency of engagement

As can be seen in Fig 1, almost half of participants reported to engage in 'financial support' on a weekly or monthly basis. 'Caring', 'emotional support', 'sharing' and 'kinkeeping' activities were reported to be carried out weekly by most participants. Over 80% of participants reported to provide guidance to their children on a weekly or monthly basis, regarding various everyday issues (such as cooking, home maintenance). Only 3.2% of participants stated that they never engage in generativity occupations.

**Table 1. The demographic information of the older women (N = 317) who participated in the survey.**

| | Mean (SD) | Min-Max |
|---|---|---|
| Age | 69.70 (4.25) | 65–87 |
| Education (years) | 16 (3.07) | 6–30 |
| Number of children | 3.01 (1.03) | 1–7 |
| | *N(%)* | |
| **Marital status** | | |
| Married or in a relationship | 231 (66.2) | |
| Never married/ Divorced/ Widowed | 4/ 44/ 38 (1.3/ 13.9/ 12.0) | |
| **Live** | | |
| Alone/ with Spouse/ Child/ other | 78/ 228/ 10/1 (24.6/71.9/3.2/.3) | |
| **Place of birth** | | |
| Israel/ Europe/ North America/ South America/ Africa/ Middle East/ Asia | 245/ 35/ 9 / 8/ 6/ 6 (77.3/ 11.0/ 2.8/ 2.5/ 1.9/ 1.9/ 0.6) | |
| **Religious status (Jewish)** | | |
| Secular / Conservative/ Religious | 241/ 51/ 25 (76.0/ 16.1/ 7.9) | |

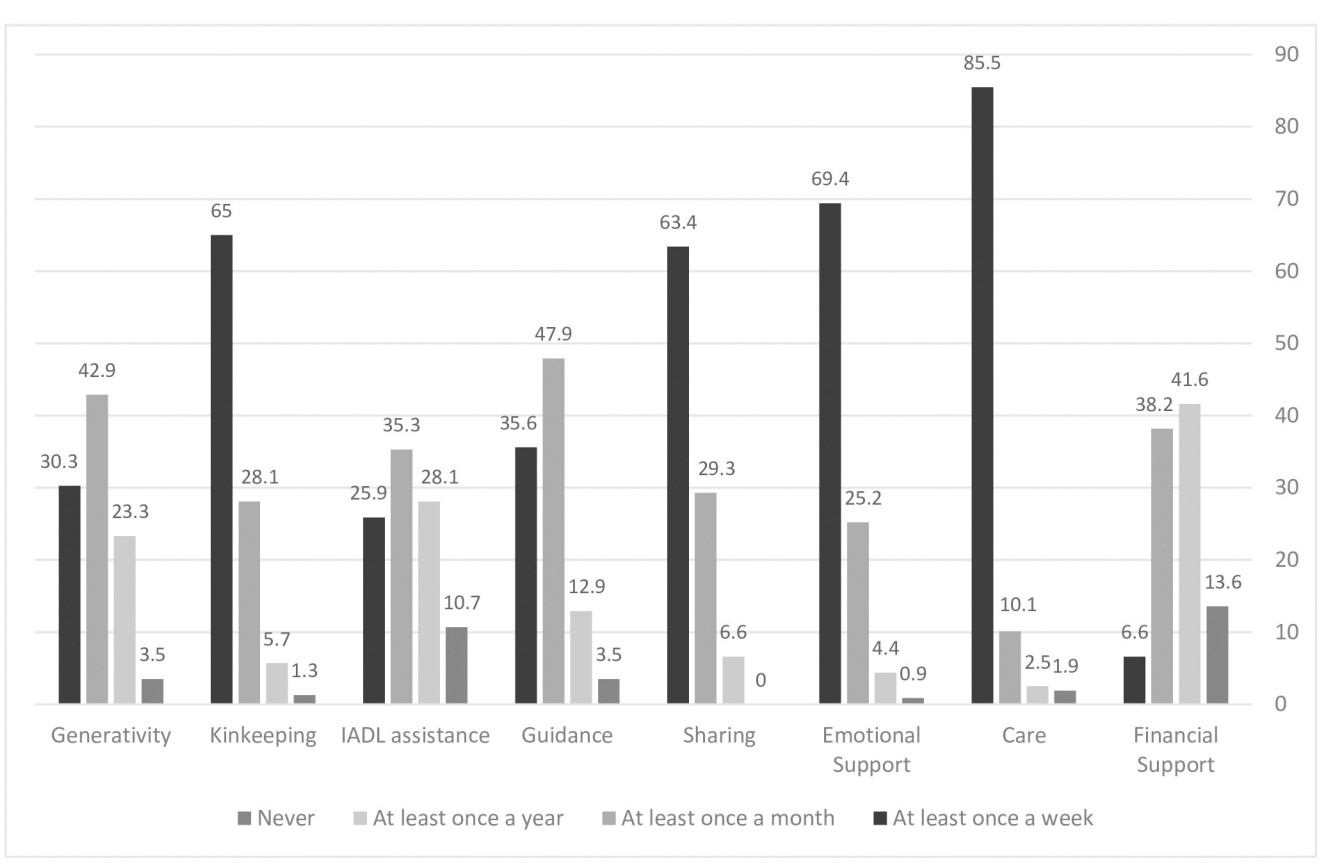

**Fig 1. Frequency of engagement in maternal occupations (N = 317); percentage of the mother participants who engage in maternal occupations at least 'once a week', 'once a month', 'once a year' or 'never'.**

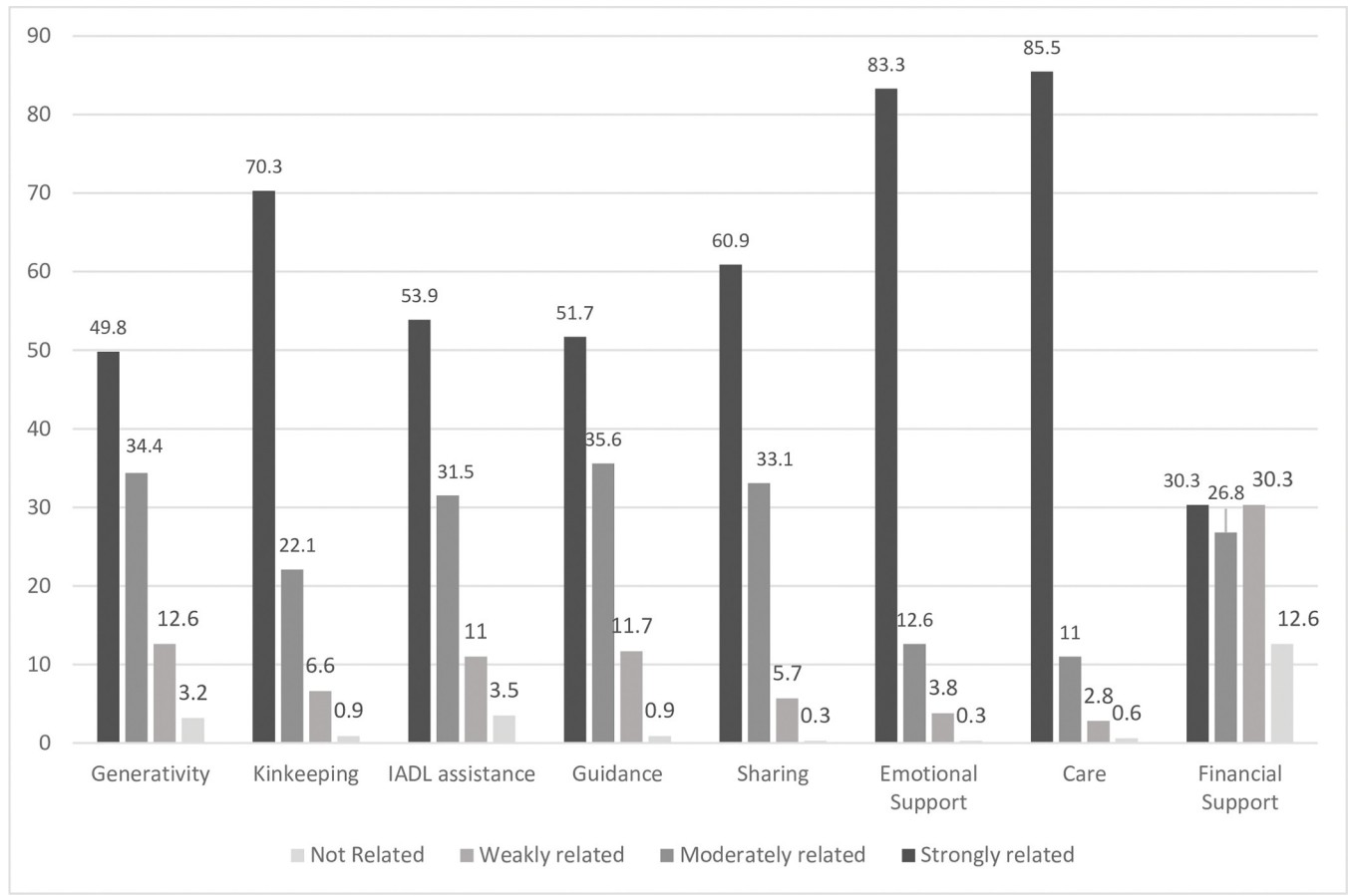

**Fig 2. Relatedness of occupations to the maternal role; percentage of the participants (N = 317) who rated the occupations as 'not related', 'weakly', 'moderately' or 'strongly related' to the maternal role.**

## Relatedness

As can be seen in Fig 2, most participants rated 'caring' and 'emotional support' as occupations strongly related to the maternal role. 'Sharing' and 'kinkeeping' (were rated strongly-related to the maternal role by more than half the participants, while 'guidance', 'IADL assistance' and 'generativity' were rated strongly related by approximately 50% of the participants. 'Financial support' was rated by approximately 50% as not related or weakly related to their maternal role.

## Perceptions

Fig 3 describes the participants' level of agreement with statements regarding the maternal role. Most mothers perceive motherhood as a life role which does not cease when children grow older and perceive that their maternal role has changed as opposed to their past. The perception that assisting children with daily chores was common among most mothers, and over 90% agreed that it is important that they are present when their children need to share their thoughts. Over 70% of the mothers perceive themselves as responsible to maintain the family structure, and over 80% perceive importance in passing on the family legacy to the next generation.

Two open ended questions were answered by 113 of the 317 participants and the thematic analysis was performed. Based on their responses, we revealed the following seven categories

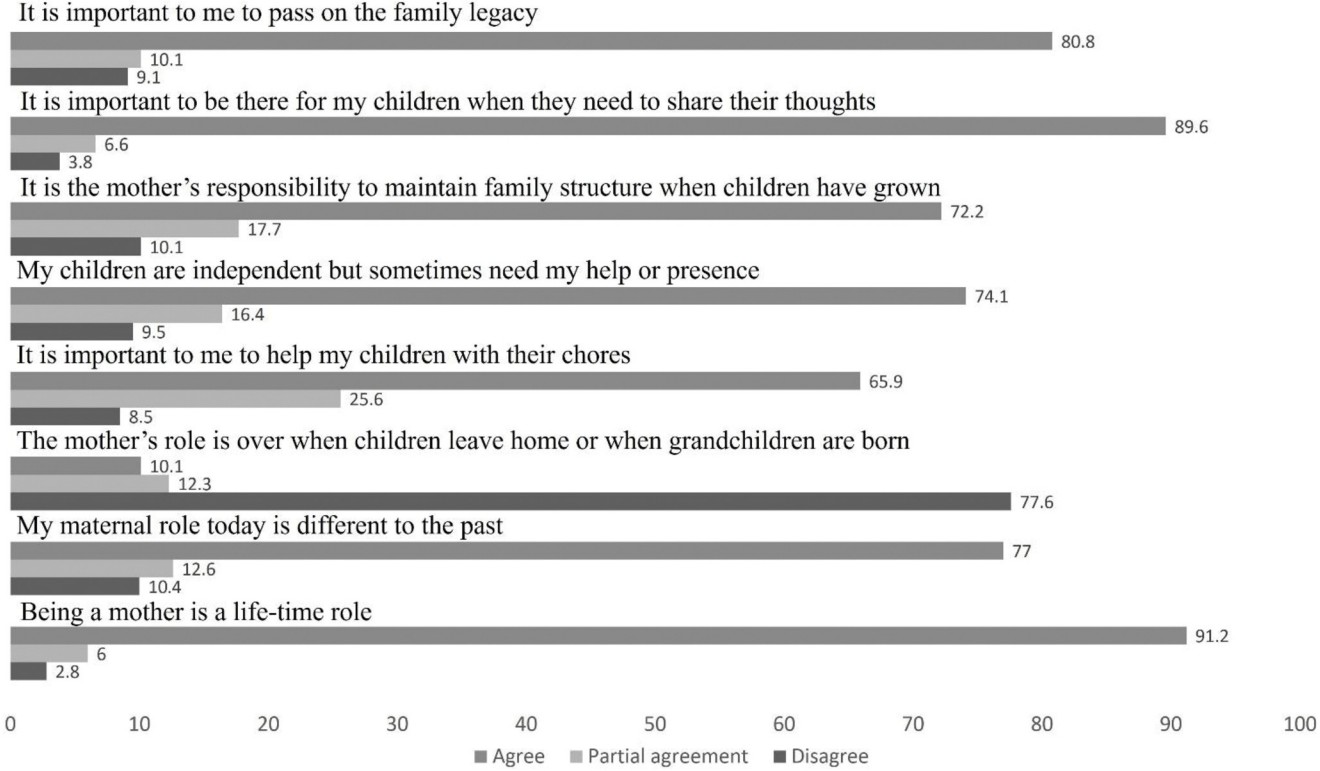

**Fig 3. The percentage of the participants (N = 317) who rated their agreement, partial agreement or disagreement with statements related to perceptions on the maternal role.**

regarding the maternal role at older age. These categories describe both 'doing' and 'being' in the maternal role and included specific activities they perform as mothers, as well as motherhood emotions such as being concerned or worrying about the children and grandchildren, thoughts and hope regarding the present and the future. The following seven categories are supplementary to the occupations that we proposed in the closed-ended questions.

### Providing instrumental support

Participants described assisting their adult children with daily chores at home and around the home. They elaborated and described concrete examples of support such as cooking, ironing, housekeeping, and babysitting the grandchildren, when needed. In addition, older mothers reported to provide financial support and technical assistance (e.g. consulting for grandchildren's academic studies). For example, participant number 154 (P154), a 67-year-old mother to two children, shared: "I assist with raising the grandchildren on a daily basis. I host [dinners] on Friday nights and provide financial assistance when needed and when possible."

### Keeping close relationships with children and grandchildren

Participants described activities carried out with their children (e.g. family trips and meetings at coffee shops). P62, 65 years old, mother of two children, described: "I made it a tradition to meet with my daughters for 'fun days and go on family trips". The mothers also described having frequent conversations with their children, and characterized these interactions as opportunities to share and honor each child as an adult. Keeping a close relationship and spending

time with grandchildren were also recognized by many participants as an inherent part of the maternal role.

### Growth and reciprocity

Mothers described their maternal role development, which changed over time. For example, they acknowledged receiving emotional or physical support from their adult children as they aged (change in the direction of support). P188, 67 years old, mother of four children, elaborated: "It is a process that changes over the life course. At first the parents are the 'educators and deciders'. As the children grow to be adults and parents, the relationship changes and is more equal until the parents grow older and then, sometimes, the children need to 'take command'. We, the older parents, need to learn to ask our children for assistance. It is hard to acknowledge that our children are mature and capable enough to manage their life independently (as we did) and that they can provide us with physical and emotional support".

### 'Being' in the maternal role

Besides 'doing' activities and tasks, the maternal role was also portrayed by 'being' aspects, for example mothers' feelings of content and pride in their children. The older mothers also described enjoyment and sense of meaning that emerged from recognizing the positive outcomes from the ongoing process of mothering. Furthermore, mothers expressed the significance in "being there" for their children; being available and present for their children. P179, 74 years old, mother of three children, expressed this idea: "The presence of an older mother as part of the family is important in its' self, regardless of the assistance that she provides".

### Differentiation between parent and child

It appears that participants were concerned with the boundaries as well as the dynamics between their own life and their adult children's lives. In this regard, they underlined the importance of being flexible and recognizing when it is appropriate to "let go" and enable their children's autonomy, as well as their own. P47, 70 years old, mother of three children, stated: "To fill life with activities and interest, as to not burden the adult children. To be an independent entity, but close and involved, if involvement is necessary."

P177, 66 years old, mother of two children, added: "To try to do as much as possible for the children, while keeping ourselves in mind".

### The older Sandwich generation

The challenging dual caregiver position of simultaneously being a mother as well as caregiving their own older parents, was mentioned by several participants. P342, 74 years old, mother of four children described: "At this stage of life, I see the situation from both sides: I am still a caregiver to my older mother, and on the other hand I assist my children and 13 grandchildren".

### Hope and thoughts for the future

The future was represented by the participants as hopeful and concerns. The aspiration to continue and contribute to their children as long as possible, as well as concerns regarding the future, e.g. the hope to not be a burden to their children, in case of disability. "I wish I can maintain my energy and will not require their assistance in a way that will disturb their lives" was expressed by P 280, 72 years old, mother of two children.

## Discussion

This exploratory survey aimed to identify occupations and daily activities that older mothers ascribe to their maternal role at older age. The study followed a scoping review, that we conducted, of the occupations within the maternal role at older age, which highlighted the paucity of research on this important subject and helped formulate the questions and statements [32]. In this study, the older mothers rated a wide range of activities and occupations that are related to their maternal role, emphasizing that the maternal role is very significant also when the children are grown-up, are independent and have families of their own.

The 317 mothers who answered the online survey were all 65 years old or older, which is older compared to earlier studies that included older mothers [For example: 17, 43–45]. This sample was heterogeneous in their socio-demographic characteristics (such as age of mothers, number of children, marital and religious status), which is an advantage, because these factors were previously found to impact various aspects of motherhood [46–48].

### Frequency of engagement and relatedness of occupations to the maternal role

The strong relatedness of 'assisting in IADL', 'guidance' and 'sharing' to the maternal role supports previous research on motherhood of younger [28, 38] and older mothers [32]. However, the relatedness of 'caring', 'kinkeeping' and 'generativity' to the maternal role, indicated by the older participants, have not been previously identified as motherhood occupations. The high frequency of engagement reported for these occupations, indicates that these occupations are central within the maternal role of older women.

Caring for children is an essential part of motherhood [19], but this term mostly relates to caregiving of young children who are dependent in basic activities of daily living [38, 49]. Since older mothers are not typically required to this kind of care, we conceptualized 'caring' as 'being concerned with the wellbeing of children'. Naturally, the engagement in this occupation may not be visible to others, and may be linked to 'being'; as opposed to 'assisting with IADL' and 'guidance' occupations (such as cooking, cleaning or giving household advice) which can be easily linked to 'doing' within the maternal role. Since 'being' aspects of caring within the maternal role of older women have not been widely explored, our findings suggest a need for further research.

The high relatedness and frequency of engagement in kinkeeping and generativity occupations, hints to the need to explore the relationship of these occupations with the health and wellbeing of older mothers. Kinkeeping occupations are aimed towards keeping family members in touch with each other [39], and generativity activities aim to contribute to the family or community in order to ensure the continuity between generations [40]. Both have been found to be beneficial for the health and wellbeing of older adults [50–53]; but have not been studied particularly within the context of the maternal role.

It is important to point out the variance in frequency and relatedness of engagement in some of the maternal role occupations. For example, only a third of the participants perceived financial support to be strongly related, while more than a third of the participants rated it to be weakly related or not related at all to their maternal role. Interestingly, 12.6% of the older mothers reported no relatedness at all. This may be explained by our diverse population in terms of personal and socio-demographic characteristics which have been found to be associated with providence of financial support by older mothers towards their adult children [54, 55]. These finding highlight the importance of understanding the subjective aspects (culture, education, socio-economical) of older mothers, which may impact their occupational choices.

## Perceptions towards the maternal role at older age

The statement 'Being a mother is a life-time role' was supported by almost all the participants. This remarkable consensus is in accordance with previous studies of mothers that related to the 'never-ending' nature of motherhood [19, 56]. However, most mothers agreed that their maternal role at older age is different compared to the past, which supports the need to understand mothering occupations in different life stages. The other statements were also supported by most of the mothers, suggesting that the maternal role has a broad nature also at older age: older mothers perceive themselves responsible for maintaining family structure; wish to be there for their children if they need assistance with daily chores or want to share their thoughts. Therefore, the maternal role keeps older mothers very active, which may be beneficial to their health and wellbeing [57, 58], but might also pose challenges to their daily schedule.

Our qualitative findings highlight the unique challenges and attributes of the maternal role at older age. The older mothers described the challenge of being involved, while staying cautious of the autonomy of themselves and their children. Friction between parental involvement and autonomy is a psychological aspect of family dynamics at later life [59], as older parents strive to balance conflicting attitudes and perspectives regarding their parental role [60]. Our findings may hint regarding occupational strategies which older mothers use (consciously or not) to achieve this balance, for example reinforcing their engagement in occupations external to the maternal role in attempt to not interfere in their children's autonomy. The sandwich generation was described as another challenge at older age. Traditionally 'The sandwich generation' relates to working women who and have at least one child under the age of 18 but also have caring responsibilities to at least one parent or parent-in-law [61]. These women who handle multi-roles, are at higher risk of compromised health and wellbeing [62, 63]. Our findings suggest that the challenges and consequences related to the sandwich generation go beyond the stage of being employed and into older age.

Alongside the 'doing' and 'being' aspects, it seems that maternal role occupations of older women also consist of a 'becoming' dimension. The older mothers described the maternal role as a dynamic process, which constantly changes, parallel to the development of their children: from young and dependent to adult and independent. 'Becoming' is related to changes which occur across the life time and can lead to human development and growth [64]. While these changes are natural, they often require attentive responses or adjustments in order to maintain health and wellbeing [65]. The concern that mothers expressed regarding maintaining their functional abilities in the future, may hint towards the need to make further adjustments to their engagement within the maternal-role in the new stages in their life.

This calls for further research to deepen the understanding of the subjective aspects of maternal role participation. The idea that a more specific and subjective categorization of occupations is needed to describe the participation of older adults has been discussed over the past years [23, 35–37]. Our findings seem to support this notion. This information may assist healthcare practitioners in the future to promote the health and wellbeing of older women by setting functional therapeutic goals which are meaningful, and therefor may enhance motivation and compliance.

This study has a few limitations, that should be acknowledged. Using an online survey enabled distribution to a wide and diverse population, however it limited the sample to women with digital access and orientation. The generalization of our conclusions is therefore limited, and it is recommended that future studies use other methods to include older mothers without digital access. Due to minimal previous research of maternal occupations at older age, we did not use a standardized questionnaire. We composed preliminary questions and

statements, based on existing relevant literature, in attempt to gather exploratory data. In the future it is recommended to include additional information regarding the older mothers (such as socio-economic characteristics, perceptions regarding health, emotional aspects, participation in social and physical occupations), to assess the associations with perceptions and engagement in maternal role activities. We also recommend performing face-to-face assessments, including qualitative inquiry to create a profound base of knowledge regarding the maternal role of older women from various backgrounds.

To conclude, this exploratory study illuminates unique characteristics of the maternal role at older age. Some occupations which were found central in the maternal role of older women (such as generativity and kinkeeping occupations) were not previously identified as maternal occupations of younger mothers. Older mothers perceive their motherhood as an important and meaningful life role, which continues to evolve throughout time. The challenges and attributes highlighted by the older mothers, emphasize the importance to continue the research to deepen the understanding of the unique characteristics of the maternal role at older age.

## Supporting information

**S1 Data.**
(SAV)

## Author Contributions

**Conceptualization:** Ruth Maman, Debbie Rand, Michal Avrech Bar.

**Data curation:** Ruth Maman.

**Formal analysis:** Ruth Maman, Debbie Rand, Michal Avrech Bar.

**Investigation:** Ruth Maman.

**Methodology:** Ruth Maman, Debbie Rand, Michal Avrech Bar.

**Project administration:** Ruth Maman, Debbie Rand, Michal Avrech Bar.

**Supervision:** Debbie Rand, Michal Avrech Bar.

**Validation:** Debbie Rand, Michal Avrech Bar.

**Writing – original draft:** Ruth Maman.

**Writing – review & editing:** Debbie Rand, Michal Avrech Bar.

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
