## [Decision Letter · Decision Letter 0]

5 Jan 2023

PONE-D-22-28085How do older women perceive the occupations and activities within their maternal role? Findings from an exploratory surveyPLOS ONE

Dear Dr. Rand,

Thank you for submitting your manuscript to PLOS ONE. After careful consideration, we feel that it has merit but does not fully meet PLOS ONE’s publication criteria as it currently stands. Therefore, we invite you to submit a revised version of the manuscript that addresses the points raised during the review process.

We look forward to receiving your revised manuscript.

Kind regards,

Neelu Jain Gupta, Ph.D.

Academic Editor

PLOS ONE

Journal Requirements:

3. Please include your ethics statement in the Methods section of your manuscript. In the Methods section of your revised manuscript, please include the full name of the institutional review board or ethics committee that approved the protocol, the approval or permit number that was issued, and the date that approval was granted.

Additional Editor Comments (if provided):

Substantial changes are required to address the questions raised by the reviewer one and two. Please address the questions asked and revert with revised manuscript.

Reviewers' comments:

Reviewer's Responses to Questions

**Comments to the Author**

1. Is the manuscript technically sound, and do the data support the conclusions?

Reviewer #1: Yes

Reviewer #2: Partly

2. Has the statistical analysis been performed appropriately and rigorously? 

Reviewer #1: Yes

Reviewer #2: No

3. Have the authors made all data underlying the findings in their manuscript fully available?

Reviewer #1: Yes

Reviewer #2: Yes

4. Is the manuscript presented in an intelligible fashion and written in standard English?

Reviewer #1: Yes

Reviewer #2: No

5. Review Comments to the Author

Reviewer #1: 1.Despite the understanding of its development over time, there appears to be a void in the study of otherhood in later stages of women's lives. Only a few research, according to our literature search, dealt with the maternal role occupations of older, healthy women. Moms with functional disabilities or mothers of children with disabilities were the subjects of the majority of studies, however older women comprises a large group of older populations.

2. However The objective of this study was to outline the jobs, pursuits, and conceptions of motherhood held by older women. This knowledge might help elderly women's health and wellbeing in the future.It is crucial to comprehend the maternal function as a fulfilling life role that may inspire older women to engage in regular activities and professions, maintaining or improving their health and wellbeing in the process. BUT authors should have included a validated or specialized survey questionnaire which could have been more useful. However the used methodology and screened parameters seems to be well evaluated and results are interesting and useful.

2.In Fig3 : perceptions of The percentage of the participants who rated their agreement, partial agreement or disagreement with statements related to perceptions on the maternal role was very well explained but should also describehow elderly moms' perceptions of their health were affected by their close relationships with their children and grandchildren, how they assume their self worth , their engagement in social activity ? I suggest it should be more clearly explained and added in discussion part .

3. However this study has nicely reported links between the older mothers' life happiness and the help given to the kids. This shows that the mothering position, act as a meaningful life role has the potential to inspire older women who are at risk of disease and chronic conditions to engage in fulfilling careers, leading to better health outcomes.

Reviewer #2: The present manuscript entitled "How do older women perceive the occupations and activities within their maternal role? Findings from an exploratory survey" analyses various questions and responses for maternal roles.

However, the manuscript lacks the

1. data for financially independent participants

2. the dependance of the female participants on their present family

3. their physical disease status

4. the dependance of them or their children and vice versa

5. the cumulative data with results obtained for the participants.

5. the cumulative data with results obtained for the participants.

6. Error bars are missing, significance levels and data mapping with age is missing.

7. ANOVA can be used for conclusive reporting of the data.

The manuscript also needs thorough revision for language and repetitions. The efforts on clarity are required throughout the text.

6. PLOS authors have the option to publish the peer review history of their article (what does this mean?). If published, this will include your full peer review and any attached files.

Reviewer #1: No

Reviewer #2: No

---

## [Author Response · Author response to Decision Letter 0]

26 Jan 2023

We greatly appreciate the reviewers’ comments, following which we provided further details and clarification regarding our methods (including the survey, inclusion criteria and data analysis), improved the presentation of our results and added recommendations for future studies. We believe that the manuscript has remarkably improved following these changes. 

Below we responded to each of the reviewers’ comments to reflect the changes we made in the manuscript (which are marked by “track changes”). 

Reviewer #1

Despite the understanding of its development over time, there appears to be a void in the study of motherhood in later stages of women's lives. Only a few research, according to our literature search, dealt with the maternal role occupations of older, healthy women. Moms with functional disabilities or mothers of children with disabilities were the subjects of the majority of studies, however older women comprises a large group of older populations. However The objective of this study was to outline the jobs, pursuits, and conceptions of motherhood held by older women. This knowledge might help elderly women's health and wellbeing in the future. It is crucial to comprehend the maternal function as a fulfilling life role that may inspire older women to engage in regular activities and professions, maintaining or improving their health and wellbeing in the process. 

We appreciate the positive feedback. 

BUT authors should have included a validated or specialized survey questionnaire which could have been more useful. However the used methodology and screened parameters seems to be well evaluated and results are interesting and useful.

Thank you for your comment. We added a sentence in the Introduction regarding the fact that assessment tools for identifying perceptions or engagement of older women in mothering occupations, are not available: “This, together with the scarce literature of the maternal role at older age, and the fact that tools to characterize mothering occupations of older women are not available, creates vagueness regarding the occupations within the maternal role of older women.” We also acknowledged this as a limitation in the discussion section: “Due to minimal previous research of maternal occupations at older age, we did not use a standardized questionnaire. We composed preliminary questions and statements, based on existing relevant literature, in attempt to gather exploratory data.” 

We composed the questions of the survey based on previous literature. We are happy to share that based on the findings of the current study, we are currently developing a questionnaire to identify participation of older mothers in meaningful occupations within the maternal role.

In Fig3: perceptions of the percentage of the participants who rated their agreement, partial agreement or disagreement with statements related to perceptions on the maternal role was very well explained but should also describe how elderly moms' perceptions of their health were affected by their close relationships with their children and grandchildren, how they assume their self worth , their engagement in social activity ? I suggest it should be more clearly explained and added in discussion part . However this study has nicely reported links between the older mothers' life happiness and the help given to the kids. This shows that the mothering position, act as a meaningful life role has the potential to inspire older women who are at risk of disease and chronic conditions to engage in fulfilling careers, leading to better health outcomes.

We agree that these are important topics in relation to older mothers, unfortunately we did not collect data regarding participants' health perception, self worth and engagement in social activities. We added these significant directions of research to the recommendations for further research: "In the future it is recommended to include additional information regarding older mothers (such as economic characteristics, perceptions regarding health, emotional aspects, participation in social and physical occupations), to assess the associations with perceptions and engagement in maternal role activities.”

Reviewer #2

the manuscript lacks the

1. data for financially independent participants

2. the dependance of the female participants on their present family

Thank you for your comment. The inclusion criteria were women who independently live in the community and we did not collect data regarding participants' financial independence or dependence on their families. We added this important point to the recommendations for further studies: "In the future it is recommended to include additional information regarding older mothers (such as economic characteristics, perceptions regarding health, emotional aspects, participation in social and physical occupations), to assess the associations with perceptions and engagement in maternal role activities.”

3. their physical disease status

4. the dependance of them or their children and vice versa

The inclusion criteria were women who independently live in the community, which is suitable for indicating the health status in this population, since older adults with major diseases frequently depend on others assistance in daily living activities. Following the reviewer’s comment we added the following to the introduction sections of the revised manuscript: “This information of healthy and independent older mothers may be used in the future to develop health promotion or intervention programs that are tailored specifically for older mothers with functional decline.”

5. the cumulative data with results obtained for the participants.

We appreciate you pointing this out. We added the cumulative data obtained for the participants for each of the three figures in the revised figure attachment.

6. Error bars are missing, significance levels and data mapping with age is missing.

All of the figures present descriptive statistics (frequencies), therefore error bars and significance are not presented. Because the topic of this study has yet to be systematically explored, we chose an exploratory approach using descriptive statistics and did not explore correlations with other variables such as age.

7. ANOVA can be used for conclusive reporting of the data.

Thank you for this suggestion. as mentioned in this study we aimed to preliminarily characterize the maternal role at older age and therefore did not divide the sample into subgroups, however we added this as a recommendation for further research: "In the future it is recommended to include additional information regarding the older mothers (such as socio-economic characteristics, perceptions regarding health, emotional aspects, participation in social and physical occupations), to assess the associations with perceptions and engagement in maternal role activities.

The manuscript also needs thorough revision for language and repetitions. The efforts on clarity are required throughout the text.

We thoroughly reviewed the manuscript and made various changes to improve clarity and conciseness.

---

## [Decision Letter · Decision Letter 1]

21 Mar 2023

How do older women perceive the occupations and activities within their maternal role? Findings from an exploratory survey

PONE-D-22-28085R1

Dear Dr. Rand,

We’re pleased to inform you that your manuscript has been judged scientifically suitable for publication and will be formally accepted for publication once it meets all outstanding technical requirements.

Kind regards,

Neelu Jain Gupta, Ph.D.

Academic Editor

PLOS ONE

Additional Editor Comments (optional):

Reviewers' comments:

Reviewer's Responses to Questions

**Comments to the Author**

1. If the authors have adequately addressed your comments raised in a previous round of review and you feel that this manuscript is now acceptable for publication, you may indicate that here to bypass the “Comments to the Author” section, enter your conflict of interest statement in the “Confidential to Editor” section, and submit your "Accept" recommendation.

Reviewer #1: All comments have been addressed

2. Is the manuscript technically sound, and do the data support the conclusions?

Reviewer #1: Yes

3. Has the statistical analysis been performed appropriately and rigorously? 

Reviewer #1: Yes

4. Have the authors made all data underlying the findings in their manuscript fully available?

Reviewer #1: Yes

5. Is the manuscript presented in an intelligible fashion and written in standard English?

Reviewer #1: Yes

6. Review Comments to the Author

Reviewer #1: Authors have adequately addressed the review comments of thge last set of review.Authors have exquisitely addressed the role of working older mother role survey in manuscript now.

7. PLOS authors have the option to publish the peer review history of their article (what does this mean?). If published, this will include your full peer review and any attached files.

Reviewer #1: No

---

## [Editor Report · Acceptance letter]

29 Mar 2023

PONE-D-22-28085R1 

How do older women perceive the occupations and activities within their maternal role? Findings from an exploratory survey 

Dear Dr. Rand:

I'm pleased to inform you that your manuscript has been deemed suitable for publication in PLOS ONE. Congratulations! Your manuscript is now with our production department. 

Kind regards, 

on behalf of

Dr. Neelu Jain Gupta 

Academic Editor

PLOS ONE